# FRAGILE GIANTS: UNDERSTANDING THE SUSCEPTIBILITY OF MODELS TO SUBPOPULATION ATTACKS

## ABSTRACT

As machine learning models become increasingly complex, concerns about their robustness and trustworthiness have become more pressing. A critical vulnerability of these models is data poisoning attacks, where adversaries deliberately alter training data to degrade model performance. One particularly stealthy form of these attacks is subpopulation poisoning, which targets distinct subgroups within a dataset while leaving overall performance largely intact. The ability of these attacks to generalize within subpopulations poses a significant risk in real-world settings, as they can be exploited to harm marginalized or underrepresented groups within the dataset. In this work, we investigate how model complexity influences susceptibility to subpopulation poisoning attacks. We introduce a theoretical framework that explains how models with locally dependent learning behavior—a characteristic exhibited by overparameterized models—can misclassify arbitrary subpopulations. To validate our theory, we conduct extensive experiments on large-scale image and text datasets using popular model architectures. Our results show a clear trend: models with more parameters are significantly more vulnerable to subpopulation poisoning. Moreover, we find that attacks on smaller, human-interpretable subgroups often go undetected by these models. These results highlight the need to develop defenses that specifically address subpopulation vulnerabilities.

## 1 INTRODUCTION

Machine Learning (ML) models are increasingly adopted across a wide range of domains and industries. This progress can be attributed to the availability of large-scale representative datasets and the increasing complexity and scale of ML models (Pugliese et al.). As ML transitions from research to real-world applications, it raises significant concerns regarding privacy, accountability, and trustworthiness. Despite recent advancements, these models remain vulnerable to real-world issues, including biases and insufficient safety measures (Hendrycks et al. (2021); Amodei et al. (2016)).

In addition to the inherent brittleness of ML models, their adoption in critical applications has made them a target of adversarial attacks. One prominent such attack is data poisoning, where adversaries deliberately manipulate the training data to degrade model performance (Kumar et al. (2020); Biggio et al. (2012)). Data poisoning attacks can be classified as either untargeted, aiming to impact the model's overall functionality (Luo et al. (2023); Mallah et al. (2023)), or targeted, where specific malicious patterns are introduced to influence only a particular subset of the data (Geiping et al. (2021); Shafahi et al. (2018); Huang et al. (2020)). The expansive and frequently unverified sources of datasets used in modern ML systems offer a realistic attack surface. Traditional defenses, such as data cleaning, are often inadequate, particularly when adversarial alterations are subtle or or when the data is contributed in a concealed or encrypted way for privacy reasons (Lycklama et al. (2023; 2024)).

Subpopulation poisoning attacks present a particularly concerning attack vector in ML settings involving large, diverse datasets. In these attacks, adversaries manipulate data to degrade performance on specific subpopulations, while maintaining the model's overall accuracy on the rest of the dataset (Jagielski et al. (2021)). Unlike conventional targeted attacks, subpopulation attacks do not require access to specific test instances (Shafahi et al. (2018); Geiping et al. (2021)), making them an attractive option in real-world scenarios. Moreover, the ability of these attacks to generalize to an

entire subpopulation poses a significant risk in real-world settings, as these can be exploited to harm marginalized or underrepresented groups within the dataset.

Previous studies have shown that ML models often treat subpopulations differently, amplifying the risk that certain groups may be more susceptible to adversarial attacks. This discrepancy in model behavior arises from the nature of the structure of modern datasets, which frequently follow long-tailed distributions, with underrepresented subpopulations comprising the tail (Feldman (2020)). Modern overparameterized deep learning models, with their high capacity, are inherently capable of memorizing rare data samples from the tail of these distributions (Zhang et al. (2021)). The requirement for ML algorithms to memorize in order to perform well on common deep learning tasks has largely been studied in the context of privacy and fairness (Hooker et al. (2020b;a); Carlini et al. (2019b)). However, it may also have significant implications for robustness. As model capacity increases, so does the tendency to memorize subpopulations, which might lead to inconsistent handling and greater vulnerability to exploitation. In this work, we shed light on how variations in model capacity – and by extension, model complexity – affect the susceptibility of subpopulations to poisoning attacks.

**Contributions.** As we continue to pursue better performance, the shift toward more complex models becomes inevitable. It is, therefore, crucial to understand the trade-offs that come with this added complexity. In this work, we examine the relationship between model capacity and subpopulation poisoning attacks to better understand how robustness is affected as models become more complex. Our goal is to identify which subpopulations are most vulnerable in this context, paving the way for the development of future targeted defenses.

Specifically, we make the following contributions:

1. We highlight the vulnerability of locally-dependent mixture learners to subpopulation poisoning attacks in a theoretical framework, building on existing work regarding the memorization of long-tailed data distributions. We show why defending against these attacks can be particularly challenging.

2. We demonstrate that complex models experience greater shifts in their decision boundaries when subjected to subpopulation poisoning attacks.

3. We empirically investigate the vulnerability of realistic, overparameterized models to subpopulation poisoning attacks across real-world image and text datasets. Through 1626 individual poisoning experiments across different subpopulations, datasets, and models, we demonstrate that larger, more complex models are significantly more prone to such attacks.

## 2 RELATED WORK

In this section, we first provide an overview of related research into the analysis of subpopulations in ML. We then review existing work on subpopulation poisoning attacks. Finally, we discuss recent work investigating the connection between model capacity and poisoning attacks.

**Subpopulation Analysis.** The treatment of subpopulations in machine learning has been explored across various domains, including fairness (Ganesh et al. (2023)), privacy (Bagdasaryan & Shmatikov (2019)), learning dynamics (Mangalam & Prabhu (2019)), and adversarial robustness, particularly in test-time adversarial attacks (Raina & Gales (2023)). For instance, Carlini et al. (2019a) characterize data distributions in terms of prototypical versus rare samples across five scoring metrics related to adversarial robustness, privacy and difficulty of learning. Their analysis shows a strong correlation between these metrics for the majority of training data, suggesting the presence of a broader concept of "well-representedness" that encompasses various dimensions of data characterization. Several works have specifically examined how model capacity affects the treatment of long-tail samples. Hooker et al. (2020a) demonstrate that model pruning techniques disproportionately impact outlier data points while preserving overall model accuracy. Their results indicate that the degree of disproportionate impact increases with more aggressive pruning. Similarly, Hooker et al. (2020b) and Hooker et al. (2020a) confirm that quantized models exhibit a similar pattern of disparate treatment toward rare samples. While these studies primarily focus on individual sample-level characterizations, they strongly suggest that model capacity plays a critical role in the classification performance on long-tail distributions.

**Subpopulation Poisoning.** Jagielski et al. (2021) first formalized the notion of a "subpopulation data poisoning attack", and proposed two ways to define subpopulations, one based on data annotations and another using clustering techniques. They show that different subpopulations experience varying levels of accuracy degradation when subjected to poisoning. However, the work did not explore potential causes for this disparity among subpopulations or investigate the influence of model characteristics, such as size or complexity. Building on this foundation, Rose et al. (2023) analyze subpopulation susceptibility by visualizing decision boundaries, with a particular focus on the Adult dataset. Notably, they found no correlation between subpopulation size and susceptibility to attack, highlighting the difficulty in generalizing semantically meaningful properties that could predict subgroup vulnerability. While this study provides a deeper understanding, it is limited to linear SVM models and does not extend to more complex architectures or non-convex learning objectives. As a result, the potential role of overparameterization in subpopulation poisoning attacks, which could be significant, remains unexplored.

**Model Capacity and Poisoning.** More recently, several works have investigated the impact of model capacity on backdoor poisoning attacks, which involve hidden model behaviors triggered by specific inputs, such as an (artificial) image pattern or text phrase. For instance, Wan et al. (2023) explored the feasibility of poisoning foundational language models during instruction tuning. In their analysis, they found that larger models tend to be more susceptible to such poisoning attempts. Bowen et al. (2024) conducted a more detailed study in the direction of model capacity, examining the vulnerability of various model architectures with varying capacities across three distinct poisoning scenarios. Their results showed a general trend: larger models are more vulnerable to backdoor poisoning across all settings, with the notable exception of the Gemma-2 family of models. While these studies offer valuable insights into the relationship between model size and vulnerability to backdoor attacks, they focus specifically on single-trigger backdoors. These findings do not necessarily extend to subpopulation poisoning, which targets entire subgroups of data rather than specific triggers. Understanding how subpopulations, which may encompass broader and more diverse segments of data, interact with model architecture remains a critical and largely unexplored area.

## 3 BACKGROUND

We first outline relevant theory to model long-tailed data distributions and then discuss background on data poisoning attacks.

**Preliminaries.** We consider binary classification. Let $\mathcal{X}$ be the feature space and $\mathcal{Y} = \{0, 1\}$ be the label space. The goal is to learn a hypothesis $f \colon \mathcal{X} \to \mathcal{Y}$ that minimizes the error on some (unknown) distribution $\mathcal{D}$ over $\mathcal{X} \times \mathcal{Y}$. The model trainer samples an ordered $n$-tuple of i.i.d. samples $D = ((x_1, y_1), \ldots, (x_n, y_n))$ from $\mathcal{D}$. Given a dataset $D$ and a hypothesis space $\mathcal{H}$, a learning algorithm $\mathcal{A}$ produces a hypothesis $h \leftarrow \mathcal{A}(D)$. Typically, the learner $\mathcal{A}$ chooses $h$ via empirical loss minimization under a cross-entropy loss.

For a distribution $\mathcal{D}$ over a domain $\mathcal{Z}$, we use $z \sim \mathcal{D}$ to denote sampling $z$ from $\mathcal{D}$. For a function $F$ with domain $\mathcal{X}$, we denote by $\mathrm{D}_{x \sim \mathcal{D}}[F(x)]$ the probability distribution of $F(x)$, when $x \sim \mathcal{D}$.

### 3.1 LONG-TAILED DISTRIBUTIONS

Informally, "long-tailed" distributions are characterized by a large number of rare, atypical observations forming the "long tail". For instance, image datasets might contain objects captured from unusual angles, and word frequencies in natural language often follow Zipf's law. Learning from such distributions poses challenges for fairness and privacy, because models might produce biased or inaccurate predictions on underrepresented groups, and rare events can be used to uniquely identify individuals.

Feldman (2020) formalizes this idea by modeling the data distribution as a mixture of distinct subpopulations $\mathcal{D}_1, \ldots, \mathcal{D}_k$. The marginal distribution $\mathcal{D}(x)$ is composed of a weighted sum of component distributions $\sum_{i=1}^{k} \gamma_i \mathcal{D}_i(x)$ over the subpopulations. The mixture weights $\{\gamma_i\}_{i=1}^{k}$ determine the frequency of each subpopulation in the dataset and define the long-tail explicitly. Jagielski et al. (2021) provide a definition of such a mixture distribution based on a simplification of the model presented by Feldman.

**Definition 1** (Noisy $k$-Subpopulation Mixture Distribution (Jagielski et al. (2021))). *A noisy $k$-subpopulation mixture distribution $\mathcal{D} = \sum_i \gamma_i \mathcal{D}_i$ over $\mathcal{X} \times \mathcal{Y}$ consists of $k$ subpopulations $\{\mathcal{D}_i\}_{i=1}^k$, with distinct, known, supports over $\mathcal{X}$, (unknown) mixture weights $\gamma_1, \ldots, \gamma_k$ subject to $\sum_i \gamma_i = 1$, and labels drawn from subpopulation-specific Bernoulli distributions with probabilities $p_1, \ldots, p_k$.*

In this model, subpopulations have distinct supports, meaning that each data point is associated with only one subpopulation. With a slight abuse of notation, we denote the distribution of the subpopulation to which $x$ belongs as $\mathcal{D}_x$. Feldman demonstrates theoretically that machine learning models must "memorize" the long-tail subpopulations to minimize generalization error. This memorization is necessary because many subpopulations are represented by only a single data point in the dataset. For the model to generalize well, it needs to capture these rare instances, despite their infrequency. Building on this theoretical insight, Feldman & Zhang (2020) empirically observe this memorization behavior in real-world datasets. By identifying closely related pairs of individual training and test samples, the work highlights how models learn specific behavior of these rare instances to accurately predict rare subpopulations.

### 3.2 DATA POISONING

In a data poisoning attack, the adversary modifies the training dataset with the goal of influencing the model's behavior. Poisoning attacks can be categorized based on their objectives and the impact on the compromised model. Availability attacks aim to reduce the overall accuracy of the model across the data distribution (Biggio et al. (2012); Jagielski et al. (2018); Xiao et al. (2015); Koh et al. (2022)). In contrast, targeted attacks seek to misclassify a specific set of points, $D_{\text{target}}$, while maintaining high accuracy on the rest of the data distribution to avoid detection (Geiping et al. (2021), Huang et al. (2020), Koh & Liang (2017), Shafahi et al. (2018)). A prominent example of targeted attacks is backdoor attacks (Gu et al. (2019)), which induce misclassification by embedding an attacker-chosen trigger into test data points.

Subpopulation poisoning attacks interpolate between availability and targeted attacks (Jagielski et al. (2021)). The adversary seeks to manipulate the model's performance over an entire subpopulation, rather than specific, known instances. Unlike targeted attacks, where the adversary knows the precise set of target samples, $D_{\text{target}}$, subpopulation poisoning attacks are broader in scope, where the adversary targets a distribution, $\mathcal{D}_p$, instead of isolated points. The adversary's goal is to minimize the model's accuracy on the subpopulation without affecting predictions on points outside the subpopulation. In this work, we focus on adversaries restricted to modifying (i.e., "flipping") the labels of a subset of points within the training dataset.

**Definition 2** (Label Flipping Subpopulation Poisoning Attack). *Let $\mathcal{A}$ be a learning algorithm, let $D = \{(x_i, y_i)\}$ be a dataset and let $F : \mathcal{X} \rightarrow \{0, 1\}$ be a filter function that defines a subpopulation. A label flipping subpopulation poisoning attack adapts the training dataset through some transformation function $P : (\mathcal{X} \times \mathcal{Y}) \rightarrow (\mathcal{X} \times \mathcal{Y})$ in order to maximize*

$$\mathbb{E}_{(x,y)\sim\mathcal{D}, f\sim\mathcal{A}(D), f_p\sim\mathcal{A}(P(D))} \left[ \mathbb{I}\left[f(x) = y\right] - \mathbb{I}\left[f_p(x) = y\right] \mid F(x) = 1 \right]$$

*while minimizing the accuracy decrease for $F(x) = 0$, where $f$ is the clean model and $f_p$ represents the poisoned model. For the transformation function $P$, the adversary selects a subset of indices $S_p$ of $D$ for which it inverts the label to create the dataset $\{(x_i, 1 - y_i) | i \in S_p\} \cap \{(x_i, y_i) | i \notin S_p\}$.*

Jagielski et al propose two methods for defining the filter function $F$ that identifies the target subpopulation. The first method clusters the data based on the samples' latent space representations, leveraging the model's internal structure to define the subpopulation. This approach has been shown to achieve higher attack effectiveness, as it exploits the model's learned feature space to define closeness of samples. The second method is based on predefined semantic annotations associated with the samples in the dataset. Although more challenging to poison, semantic subpopulations are better aligned with real-world attacker objectives, which are based on meaningful domain-specific properties of the data.

## 4 EXPERIMENTAL DESIGN

We hypothesize that increasing model complexity exacerbates vulnerability to subpopulation poisoning attacks. Specifically, as models grow larger, their ability to memorize increases, resulting

in more locally dependent behavior for subpopulations, increasing the effectiveness of poisoning attacks. We begin by illustrating this local dependence behavior for subpopulation poisoning attacks in a theoretical model and highlight inherent challenges in defending against them. We then describe the methodology used in our empirical evaluation.

### 4.1 Poisoning Mixture Learners

Our goal in this section is to highlight how models with local dependence on subpopulations become increasingly susceptible to such attacks as their complexity grows. ML models exhibit varying degrees of local dependence on the subpopulations in their training data. That is, their predictions for any point $x$ are closely tied to how they generalize across the subpopulation $\mathcal{D}_x$ of $x$. We capture this dependence in a *mixture learner* that learns a classifier $f$ based on a dataset $D$ sampled from a mixture distribution. Let $\mathcal{D}$ be a noisy $k$-subpopulation mixture distribution $\mathcal{D}$ over $\mathcal{X} \times \mathcal{Y}$ as in Definition 1 consisting of $k$ subpopulations $\mathcal{D}_1, \ldots, \mathcal{D}_k$. A mixture learner is an algorithm $\mathcal{A}$ that takes as input a dataset $D$ sampled from $\mathcal{D}$ and returns a classifier $f : \mathcal{X} \to \{0, 1\}$. We assume that for every point $x$ in a dataset $D$, the distribution over predictions $f(x)$ for a random predictor output by $\mathcal{A}(D)$ is close to the distribution over predictions that $\mathcal{A}$ produces over the entire subpopulation to which $x$ belongs. This assumption of local dependence – where the learner makes similar predictions for all points in a subpopulation – forms the foundation for understanding why such models are susceptible to poisoning attacks.

**Definition 3** ($\delta$-local Subpopulation Mixture Learner). *A subpopulation mixture learner $\mathcal{A}$ takes as input a dataset $D$ of size $n$ of a noisy $k$-subpopulation mixture distribution $\mathcal{D}$, and returns a classifier $f : \mathcal{X} \to \{0, 1\}$. The learner $\mathcal{A}$ is $\delta$-subpopulation coupled if for any dataset $D \in (\mathcal{X} \times \mathcal{Y})^n$ and point $x \in D$,*

$$\mathsf{TV}\left(\mathsf{D}_{f \sim A(D)}[f(x)], \mathsf{D}_{x' \sim \mathcal{D}_x, f \sim A(D)}[f(x')]\right) \leq \delta.$$

*where* $\mathsf{TV}$ *denotes the total variation distance between the distributions.*

If $\delta$ is small, the classifier's predictions on any sample $x$ are strongly influenced by the behavior on the rest of the subpopulation $\mathcal{M}_x$. Our definition is adapted from the learner concept presented in Definition 3.1 in Feldman (2020) in the context of exploring memorization effects in machine learning models. Additionally, it generalizes the locally dependent $k$-subpopulation mixture learner (Definition A.2 in Jagielski et al. (2021)). For $\delta = 0$, we recover their definition, in which the learner exhibits complete local dependence, i.e., its predictions are entirely determined by the local properties of the subpopulation.

Completely locally dependent learners produce classifiers that are highly vulnerable to subpopulation-targeted poisoning attacks, because the adversary can manipulate the local subpopulation and cause widespread misclassification. Jagielski et al. (2021) demonstrate that for such a learner, a successful subpopulation poisoning attack exists that can misclassify points in the smallest subpopulation with a probability greater than 1/2. We show that there exists a label flipping poisoning attack for any subpopulation $\mathcal{D}_i$ for this setting.

**Theorem 1.** *Let $\mathcal{A}$ be a $\delta$-local subpopulation mixture learner for a noisy $k$-subpopulation mixture distribution $\mathcal{D}$ consisting of $k$ subpopulations $\mathcal{D}_1, \ldots, \mathcal{D}_k$ with mixture coefficients $\gamma_1, \ldots, \gamma_k$, that the minimizes 0-1 loss in binary classification. For a dataset $D \sim \mathcal{D}$ of size $n$, $\delta = 0$ and for all $i \in [k]$, there exists a label-flipping poisoning attack on subpopulation $\mathcal{D}_i$ of size $2\gamma_i n$ that causes misclassification with probability $1 - \exp\left(-\frac{9\gamma_i n}{5}\right)$.*

The proof of Theorem 1 is based on the intuition that for $\delta = 0$, the learner that minimizes the empirical loss must choose the most common label for each subpopulation. As a result, when flipping more than half of the labels in a subpopulation, the learner is forced to choose the poisoned label for the entire subpopulation. We defer the formal proof of Theorem 1 to Appendix A.

If the learning algorithm exhibits local dependence, it will naturally be susceptible to subpopulation poisoning attacks. While the theorem specifically applies to learners that are completely locally dependent, the same intuition is likely to hold for learners that are locally dependent with a small $\delta$. This is because a small $\delta$ still implies that the predictions for a point $x$ are heavily influenced by its subpopulation, even if not perfectly so. As $\delta$ decreases, the learner's sensitivity to the structure of individual subpopulations increases, making it more vulnerable to small, targeted perturbations that can cause widespread misclassification within the affected subpopulation.

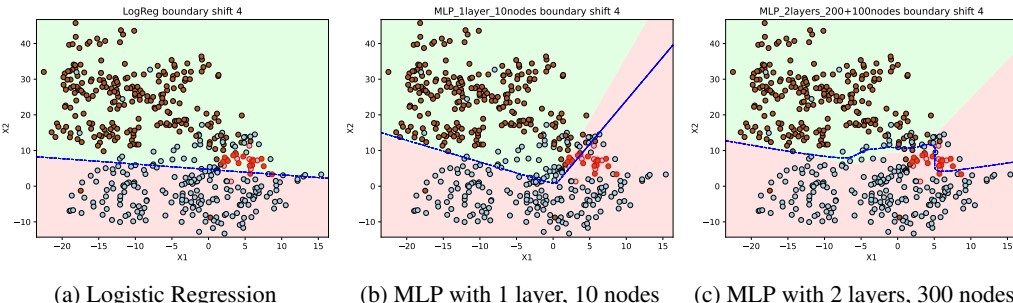

| (a) Logistic Regression | (b) MLP with 1 layer, 10 nodes | (c) MLP with 2 layers, 300 nodes |

Figure 1: Comparison of decision boundary shifts caused by a poisoning attack targeting a subgroup (red points) with $\alpha = 2.0$. The background (green-pink) represents the clean model's decision regions, while the blue line shows the boundary after the poisoning attack. The decision boundary is approximated by classifying a mesh of points across the grid.

**Implications for Overparameterized Models.** Previous work has demonstrated that local dependence is present in a variety of learning algorithms, including overparameterized linear models, $k$-nearest neighbors, mixture models, and neural networks in some settings (Li et al. (2020); Feldman (2020)). Additionally, empirical evidence shows that large, overparameterized deep neural networks tend to rely on the memorization of individual training samples to achieve low validation error on complex, realistic datasets (Feldman & Zhang (2020)). This memorization effect becomes more pronounced as model size increases, particularly for uncommon or atypical samples in the training set (Carlini et al. (2019b)). These findings suggest that larger models are more prone to local dependence on subpopulations, which in turn suggests that these are more vulnerable to subpopulation poisoning attacks.

## 4.2 METHODOLOGY

In this section, we describe the experimental framework used to measure the susceptibility of models to subpopulation poisoning attacks, detailing the subpopulations, the assumptions on the adversary, and the attack strategy. Finally, we present a toy experiment with a synthetic dataset to demonstrate the core ideas of the theoretical model and the methodology.

**Subpopulations.** We define subpopulations in our experiments using manual annotations that capture semantic information about the dataset samples, such as demographic attributes or visual characteristics in image data. These annotations, already present in the datasets we employ, represent real-world subpopulations and are commonly used in other benchmarks, such as in the context of fairness. This approach allows us to reflect realistic scenarios in which an adversary targets semantically meaningful groups based on these attributes.

We implement this concretely by defining each subpopulation through specific combinations of annotated features. Let $A = \{a_1, a_2, \ldots, a_k\}$ be the ordered tuple of binary annotation features of the dataset, and $c_1, \ldots, c_n$ denote an ordered $n$-tuple of length-$k$ binary vectors, one corresponding to each sample in the dataset $D = \{x_i, y_i\}_{i=1}^n$. For each $c_i$, the $j$-th entry in the binary vector corresponds to the presence or absence of annotation feature $a_j$. We select a subset of $m$ annotation features $A' \subseteq A$ and define the subpopulations as the cartesian product of values of $A'$. Formally, if $A'$ consists of $m$ features, then for each possible binary combination $v_k \in \{0, 1\}^m$, we define the corresponding subpopulation $D_k$ as

$$\{(x_i, y_i) \mid i \in [n] \wedge (x_i, y_i) \in D \wedge (c_i)_{A'} = v_k\}$$

where $(c_i)_{A'}$ represents the projection of the annotation vector $c_i$ onto the features in $A'$. This process yields $2^m$ disjoint subpopulations.

**Threat Model.** We assume the adversary has no access to the model weights or to the data samples in the validation set, which serves as an independent dataset used to evaluate model performance. However, the adversary is aware of the target subpopulation $\mathcal{D}_p$ within the mixture distribution to which the validation samples belong. The attacker can perturb up to $\hat{n}_p$ of the $n_p$ data samples belonging to the target subpopulation $\mathcal{D}_p$ with samples $D_p$ in the training set. This quantity $\hat{n}_p$ is defined as $\frac{(\alpha \cdot n_p)}{1 + \alpha}$ where $\alpha$ denotes the desired poisoning ratio of the subpopulation, i.e., the

number of poisoned samples in the subpopulation against the number of clean samples. For example, $\alpha = 2.0$ means double the amount of poisoned samples in the subpopulation compared to clean samples. Similar to Jagielski et al. (2021), we set the strength of the poisoning attack relative to the size of the target subpopulation in the training set.

**Attack Strategy.** The adversary performs a label flipping poisoning attack on the target subpopulation $\mathcal{D}_p$ (c.f. Definition 2). We define the filter function $F$ to return 1 if the sample belongs to the target subpopulation $D_p$ and 0 otherwise. The adversary selects $\hat{n}_p$ samples from the target subpopulation $D_p$ and changes the label of each sample from $(x_i, y_i)$ to $(x_i, 1 - y_i)$.

**Evaluation.** We train a set of models $\{\hat{f}_{D_i,\alpha}\}$, where $\hat{f}_{D_i,\alpha} \leftarrow \mathcal{A}(P_{D_i,\alpha}(D))$ with $P_{D_i,\alpha}$ representing the transformation function that applies the poisoning attack on the training set $D$ by poisoning a fraction $\alpha$ of the target samples $D_i$. We measure the effectiveness of each poisoning attack using the *target damage*, which compares the accuracy of the poisoned model $\hat{f}_{D_i,\alpha}$ to the accuracy of a clean model $f$ on validation samples from the target subpopulation $D_t \sim \mathcal{D}_i$:

$$td(f_{D_i,\alpha}) = \frac{1}{|D_t|} \sum_{(x_j,y_j) \in D_t} \mathbb{I}[f(x_j) = y_j] - \frac{1}{|D_t|} \sum_{(x_j,y_j) \in D_t} \mathbb{I}[\hat{f}_{D_i,\alpha}(x_j) = y_j]$$

**Warm-up: Gaussian Experiment.** We illustrate the local dependence behavior of models discussed in Section 4.1 in a simple example using a synthetic dataset using this methodology. We compare a series of models with varying numbers of layers and nodes on a synthetic 2D dataset with Gaussian-distributed subpopulations. We provide more details on the setup of this experiment in Appendix B. We poison a specific subpopulation in the training set and visualize the effect on the decision boundary for each model. Figure 1 displays the shift of the decision boundary for a poisoning attack on the subpopulation. The low-complexity models, such as the logistic regression model, are minimally affected by the poisoned points, with the decision boundary remaining close to the clean variant due to the support of samples of close but different subpopulations with the correct label. In contrast, the more complex models, such as the Multi-Layer Perceptron (MLP) with two layers and 300 nodes, are significantly more sensitive to the poisoning attack, with the decision boundary allowing for more flexible deviations from the clean variant to fit the poisoned subpopulation. We provide results for three additional models in Figure 6 in Appendix C.

## 5 RESULTS

We conduct experiments on three datasets: Adult, CivilComments, and CelebA. For each dataset, we use models of varying complexity and apply poisoning attacks to subpopulations defined by the datasets' annotations. We experiment with multiple poisoning ratios $\alpha \in \{0, 1, 2\}$ for Adult and CivilComments, and $\alpha \in \{0, 1, 2, 4\}$ for CelebA.

The UCI Adult dataset contains tabular data about people with loans, with subpopulations defined by ethnicity, gender, and education (Becker & Kohavi (1996)). As a simpler and lower-dimensional dataset compared to the others, we use MLPs with varying numbers of layers and widths, along with a baseline logistic regression model, which is commonly applied to this dataset. CivilComments is a dataset consisting of internet comments, with subgroups based on thematic annotations (Borkan et al. (2019)). We use four BERT models (BERT Tiny, BERT Small, BERT Medium, DistilBERT) to predict the toxicity of the comments. CelebA is a dataset of images of celebrities (Liu et al. (2015)), where the subgroups are based on age, hair color, skin tone, chubbiness, and whether the person has a beard. We use three ResNet models (ResNet18, ResNet50, ResNet101) to predict the gender of the person. We provide additional details on the datasets and their subpopulations in Appendix B.

**Model Complexity Correlates with Target Damage.** Across all datasets, we observe that models with similar architectures but greater capacity exhibit worse performance across subgroups when poisoned, as depicted in Figure 2. This effect is more pronounced as the attack strength increases: for example, at $\alpha = 1.0$, the largest models exhibit $82\%, 27\%$, and $0\%$ higher target damage than the smallest models in Adult, CivilComments, and CelebA, respectively. At $\alpha = 2.0$, these differences further increase to $89\%, 45\%$, and $22\%$, respectively. Larger models not only have higher target damage but also have lower absolute accuracy under poisoned conditions compared to the smaller models. For instance, in CivilComments, DistilBERT exhibits lower accuracy than BERT Tiny on 21 out of 32 subgroups at $\alpha = 2.0$, and in CelebA, this is the case for 6 out of 8 subgroups.

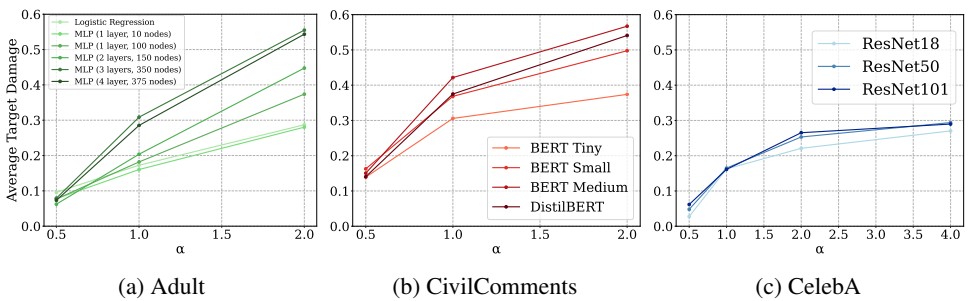

Figure 2: Average Target Damage across subgroups for increasing poisoning ratio.

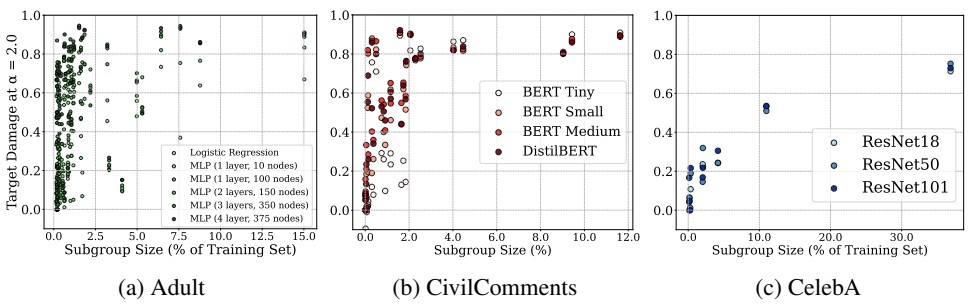

Figure 3: Relationship between subgroup size and target damage at $\alpha = 2.0$.

**Target Damage Varies Across Datasets.** We observe significant differences in attack susceptibility, highlighted by the variation in target damage among subgroups of different relative sizes across the datasets in Figure 3. For instance, for a subgroup that constitutes 3% of the training set in the Adult dataset, the largest model can experience up to 80% target damage, while in CivilComments and CelebA, the corresponding figures are around 35% and 25%, respectively. We conjecture that these variations stem from the degree to which subgroup-defining features are explicit in the training data. In Adult, the subgroup features are directly contained in the sample features; In CivilComments, subgroups are defined by the presence of a small set of specific words in the comments; while in CelebA, subgroup annotations such as hair color or age must be inferred from the images by the model. This suggests that the models might not effectively disentangle subgroup-specific features when they are implicit, leading to lower susceptibility to targeted attacks compared to datasets where subgroup features are explicit and readily accessible to the model.

## 5.1 SUBPOPULATION ANALYSIS

We now focus on the results for individual subpopulations in CelebA and CivilComments. As opposed to the models for the Adult dataset, the models used in these datasets have a high number of parameters relative to the dataset size, making them prone to overfitting on the training data. As discussed in Section 4.1, this likely has implications for their subpopulation-local behavior.

We explore the difference in accuracy for the clean and poisoned model for subgroups of different sizes. Overall, we observe a positive correlation between subgroup size and poisonability in Figures 4a and 5a. For example in CivilComments, the subgroup with size 314 has an average target damage of 48% for poisoning with $\alpha = 2.0$, whereas the subgroup with size 5541 has 79% target damage. In more detail, the results reveal a hinge-like relationship between subgroup size and target damage that is consistent across datasets. Thus, we group results across three categories of subgroups: small, mid-sized, and large, and discuss conclusions for each in more detail.

**Small Subgroups are Difficult to Poison.** There appears to be a threshold for subgroup size below which the poisoning attack does not achieve significant target damage. This behavior holds across all models and poisoning rates. Figures 4b and 5b highlight that the smallest subgroups in CivilComments and CelebA exhibit insignificant target damage (i.e., 0.04 and 0.09 on average respectively) even at higher poisoning rates. As a result, it appears that subpopulation poisoning attacks on small subgroups defined by semantic annotations is infeasible, as the model does not generalize the behavior across the entire subpopulation.

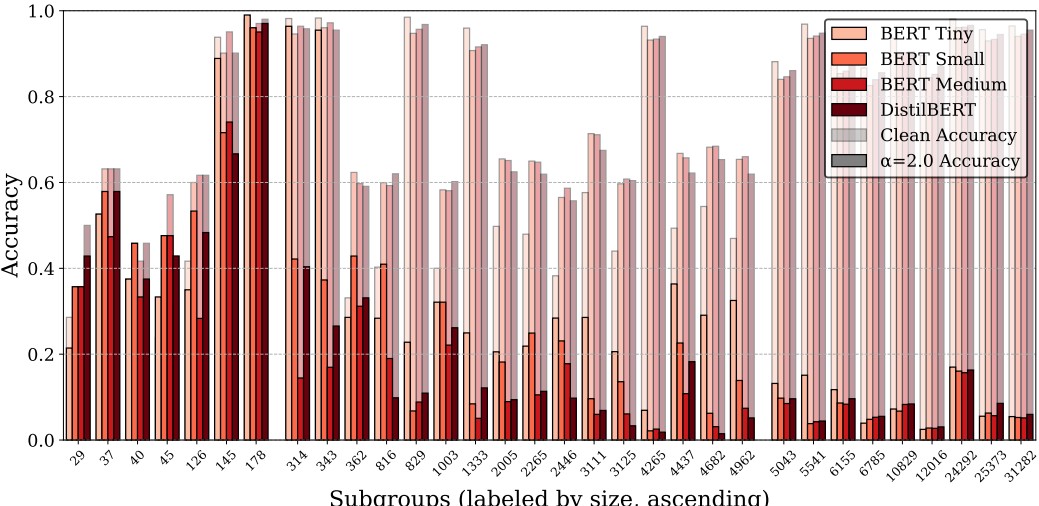

(a) Clean and poisoned accuracy per model and subgroup.

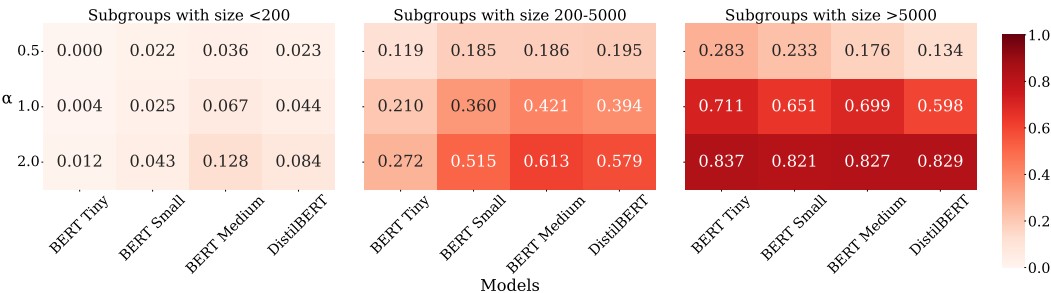

(b) Average subgroup damages across models and poisoning ratios.

Figure 4: CivilComments subgroup-level damage analysis.

**Larger Subgroups Are Less Affected by Model Size.** For larger subgroups, we observe that target damage converges across different model complexities, particularly at higher poisoning rates. In CivilComments at $\alpha = 2.0$, the difference in target damage across models is only 2%, while in CelebA, it narrows to just 1%. This trend is expected, as the poisoning attack constitutes a substantial portion of the total dataset for these larger subgroups, exerting a strong influence on the learning process. Consequently, even smaller models experience substantial shifts in their decision boundaries, despite them being more rigid, reducing the impact of model size on attack susceptibility.

**Model Complexity Affects Medium-Sized Subgroups Disproportionately.** Figures 4a and 5a additionally reveal a notable range of medium-sized subgroups where model complexity significantly impacts target damage. In CivilComments, for example, BERT Tiny exhibits only 2% target damage on a subgroup of 314 samples (subgroup "nontoxic", "buddhist"), while BERT Medium incurs 82% target damage for the same subgroup. This pattern is consistently observed across most subpopulations within this size range in CivilComments. For CelebA (cf. 5a), although this ordering by model size is less significant, we do observe a greater variance between models for the medium-sized subgroups. However, as the number of subgroups in this medium-size range is small, further investigation with a broader set of subpopulations may shed further light on this conclusion.

Finally, across both datasets, we observe a substantial variation in subgroup susceptibility to poisoning across models, independent of subgroup size. For example, the subgroup with size 3111 (subgroup "toxic", "black") in CivilComments exhibits at most 22% difference in target damage between models, which is much smaller than for subgroup 314 (80%). This suggests that some subpopulations are more prone to being memorized by models with higher capacity than others, and that these subpopulations may lie on the long tail of the distribution.

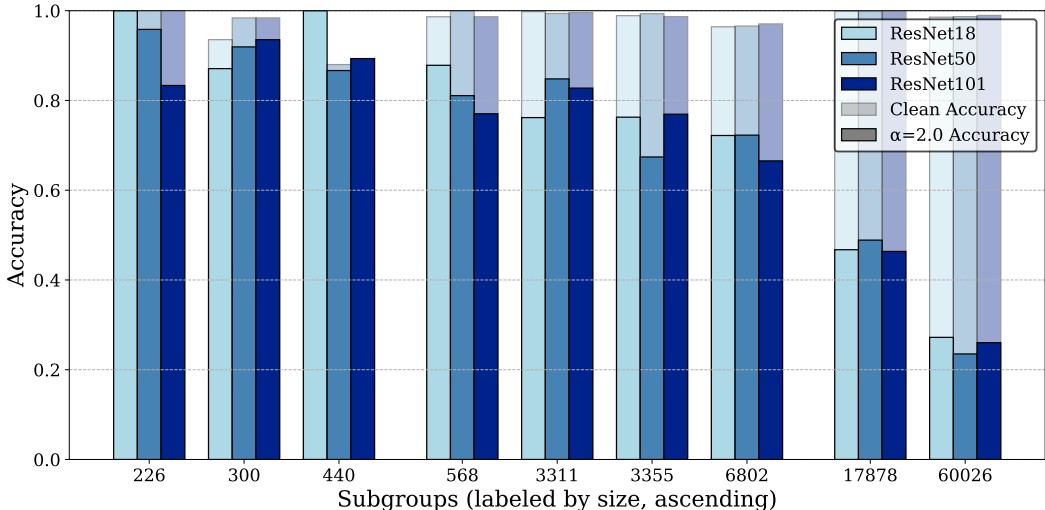

(a) Drop in subgroup accuracy on different models, grouped by behaviour.

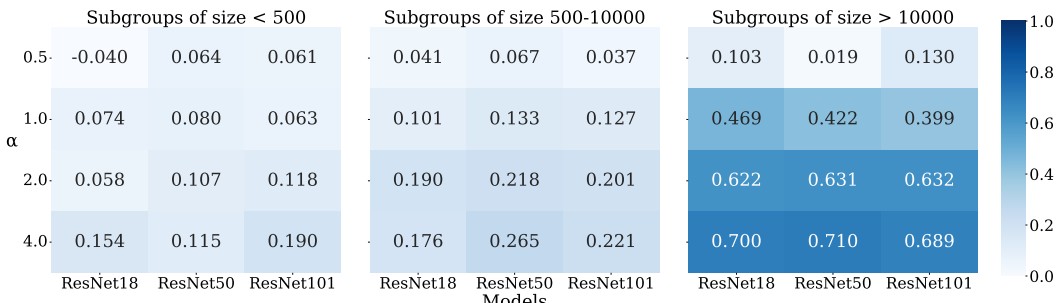

(b) Subgroup damages across models and poisoning ratios.

Figure 5: CelebA subgroup-level damage analysis.

## 6 CONCLUSION AND FUTURE WORK

This work sheds light on the intricate relationship between model complexity and the vulnerability of machine learning systems to subpopulation poisoning attacks. Our analysis highlights that as models grow in complexity, their susceptibility to such attacks increases, particularly in cases involving underrepresented subpopulations. This vulnerability is exacerbated by the long-tailed nature of modern datasets, where subpopulations may be more exposed to adversarial manipulations.

**Future Directions.** Our findings underscore the importance of considering subpopulation-specific risks in the design of defenses against poisoning attacks. The effect of learning subpopulations-specific behavior has largely been studied in the context of privacy and fairness. However, as we show in this work, it also has significant implications for robustness. Consequently, techniques to improve robustness inevitably result in tradeoffs with respect to accuracy, privacy, fairness, and robustness, and future defenses must carefully balance these objectives to achieve meaningful improvements. Moreover, our results show that attack success depends heavily on the choice of target subpopulation, highlighting the necessity of evaluating defenses across diverse subpopulations in benchmarking datasets. Finally, identifying vulnerable subpopulations can guide the development of robust training pipelines using techniques from fairness and privacy research, such as balanced sampling or re-weighting Navarro et al. (2024); Richards et al. (2023).

Our work surfaces a characteristic of long tail subgroups; future work could measure the agreement of this characteristic with similar metrics in the domains of fairness or privacy, such as group or sample level memorization (Carlini et al. (2023); Jiang et al. (2020)). This could help determine whether the observed behavior is a general property of these subpopulations or specific to poisoning, offering insights into model behavior on long-tail distributions.

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

## A  PROOF OF THEOREM 1

We provide a proof of Theorem 1 in this appendix.

*Proof.* The label-flipping poisoning attack on subpopulation $\mathcal{D}_i$ alters $2\gamma_i n$ points from $D_i$ to have the opposite label through its transformation function $P$. At a high level, the proof shows that no learner minimizing the empirical 0-1 loss can distinguish between the true labels and the flipped labels, because the 0-locally dependent learner must make a consistent decision for the entire subpopulation.

We present a proof by contradiction. Let $x$ be any sample from $D_i$ with correct label $y$, let $D' \leftarrow P_i(D)$ be the dataset after poisoning the samples from $\mathcal{D}_i$ in the dataset $D$. Suppose there exists a 0-local subpopulation mixture learner $\mathcal{A}$ that is not affected by the poisoning attack, i.e.,

$$\Pr\left[\mathcal{A}(D)(x) = \mathcal{A}(D')(x)\right] > \frac{1}{2}. \tag{1}$$

Since the learner minimizes the 0-1 loss, it must correctly classify the largest number of samples in the dataset $D$, implying that it should correctly classify the largest number of samples in $D_i$ as $D = \bigcup_j^k D_j$. For a 0-locally dependent mixture learner, this corresponds to choosing the majority label in each subgroup $D_j$. Thus, if the number of samples flipped by the poisoning attack is more than half $|D_i|$, the classifier output by the learner $\mathcal{A}(D')$ must predict the label $1 - y$. Since the original classifier $\mathcal{A}(D)(x)$ predicts the label $y$, this leads to a contradiction in Equation (1).

| SIZE | FEATURES |
|---|---|
| 226 | Blond Hair, Old, Pale Skin, No Beard, Slim |
| 300 | Dark Hair, Old, Pale Skin, No Beard, Slim |
| 440 | Dark Hair, Old, Dark Skin, No Beard, Chubby |
| 568 | Dark Hair, Young, Dark Skin, No Beard, Chubby |
| 3311 | Blond Hair, Old, Dark Skin, No Beard, Slim |
| 3355 | Dark Hair, Young, Pale Skin, No Beard, Slim |
| 6802 | Dark Hair, Old, Dark Skin, No Beard, Slim |
| 17878 | Blond Hair, Young, Dark Skin, No Beard, Slim |
| 60026 | Dark Hair, Young, Dark Skin, No Beard, Slim |

Table 1: CelebA subgroup sizes and corresponding features

We now show that $2\gamma_i n$ flipped samples are enough for a successful poisoning attack with overwhelming probability. In particular, the value $2\gamma_i n$ should be larger than $\frac{1}{2} \cdot |D_i|$, i.e., half the number of samples in the subpopulation $D_i$. The number of points in subpopulation $D_i$ is the sum of $n$ independent Bernoulli trials with $\gamma_i$, i.e., $\sum_{i=1}^{n} \text{Bernoulli}(\gamma_i)$. Applying the multiplicative Chernoff bound $\Pr[X > (1+\delta)\mu] < \exp\left(-\frac{\delta^2 \mu}{2+\delta}\right)$ with $\mu = \gamma_i n$ and $\delta = 3$, gives

$$\Pr\left[|D_i| > 4\gamma_i n\right] \leq \exp\left(-\frac{9\gamma_i n}{5}\right).$$

Hence, the size of the subpopulation $|D_i|$ is smaller than $4\gamma_i n$ with probability $1 - \exp\left(-\frac{9\gamma_i n}{5}\right)$. □

## B  MODELS, DATASETS AND SUBPOPULATIONS

We provide a detailed description of the models, datasets and subpopulations used in our experiments in this appendix.

**Gaussian Dataset.** We generate a synthetic 2D dataset using Gaussian distributions. The dataset has two classes, each composed of multiple subgroups (clusters). The distance between the class centers is given by the class separation parameter and 25 subgroup centers are then scattered around the class centers using a specified standard deviation. For each subgroup, a random number of points is generated from a normal distribution around its center, and the points are assigned the corresponding class label. Finally, a fraction of labels is randomly flipped to introduce noise, simulating label errors in the dataset.

**Adult.** UCI Adult is a tabular dataset with 32561 training samples, with the task of predicting whether a person's income is above $50K a year using the other demographic attributes. Here we form the subgroups using the features Ethnicity, Gender and Education. We filter out subgroups smaller than 100 training points, yielding 63 subgroups.

**CelebA.** The CelebFaces Attributes Dataset (Liu et al. (2015)) is an image dataset of human faces accompanied by 40 attribute annotations per image. The training set has 162,770 samples. We use the attribute 'Male/Female' as the task. We train the ResNet model family on this dataset, specifically ResNet18, ResNet50 and ResNet101, with 11 million, 26 million and 45 million trainable parameters respectively (He et al. (2016)). We select the attributes "Dark_Hair", "Young", "Pale_Skin", "Beard", "Chubby" to divide the data into subpopulations, and then filter such that the subgroups have at least 100 training samples and at least 10 test samples, resulting in 9 subgroups.

**CivilComments.** The CivilComments Dataset (Borkan et al. (2019)) is a text dataset where each entry has 19 binary annotations, such as toxic, male, Christian, etc. The task is to identify whether a given comment is toxic, while the other annotations are used to form the subgrouping together with toxicity (e.g., nontoxic male). This yields 36 subgroups in total, though four of them are too small to consider (see Table 2). In total, there are 269037 training samples and 133781 test samples. We use pre-trained versions of BERT: bert-tiny, bert-small, and bert-medium (Turc et al. (2019)) as well as DistilBERT (Sanh et al. (2019)) and added and trained a classification layer.

| SIZE | FEATURES |
|------|----------|
| 2 | toxic, other_sexual_orientation |
| 2 | toxic, other_gender |
| 4 | nontoxic, other_sexual_orientation |
| 5 | nontoxic, other_gender |
| 29 | toxic, other_religion |
| 37 | toxic, hindu |
| 40 | toxic, buddhist |
| 45 | toxic, bisexual |
| 126 | toxic, atheist |
| 145 | nontoxic, bisexual |
| 178 | nontoxic, other_religion |
| 314 | nontoxic, buddhist |
| 343 | nontoxic, hindu |
| 362 | toxic, transgender |
| 816 | toxic, jewish |
| 829 | nontoxic, atheist |
| 1003 | toxic, other_religions |
| 1333 | nontoxic, transgender |
| 2005 | toxic, homosexual_gay_or_lesbian |
| 2265 | toxic, LGBTQ |
| 2446 | toxic, christian |
| 3111 | toxic, black |
| 3125 | toxic, muslim |
| 4265 | nontoxic, jewish |
| 4437 | toxic, male |
| 4682 | toxic, white |
| 4962 | toxic, female |
| 5043 | nontoxic, homosexual_gay_or_lesbian |
| 5541 | nontoxic, other_religions |
| 6155 | nontoxic, LGBTQ |
| 6785 | nontoxic, black |
| 10829 | nontoxic, muslim |
| 12016 | nontoxic, white |
| 24292 | nontoxic, christian |
| 25373 | nontoxic, male |
| 31282 | nontoxic, female |

Table 2: CivilComments subgroup sizes and corresponding features.

## C ADDITIONAL EXPERIMENTAL RESULTS

We provide additional visualizations of the decision boundary shift for the toy Gaussian dataset in Figure 1.

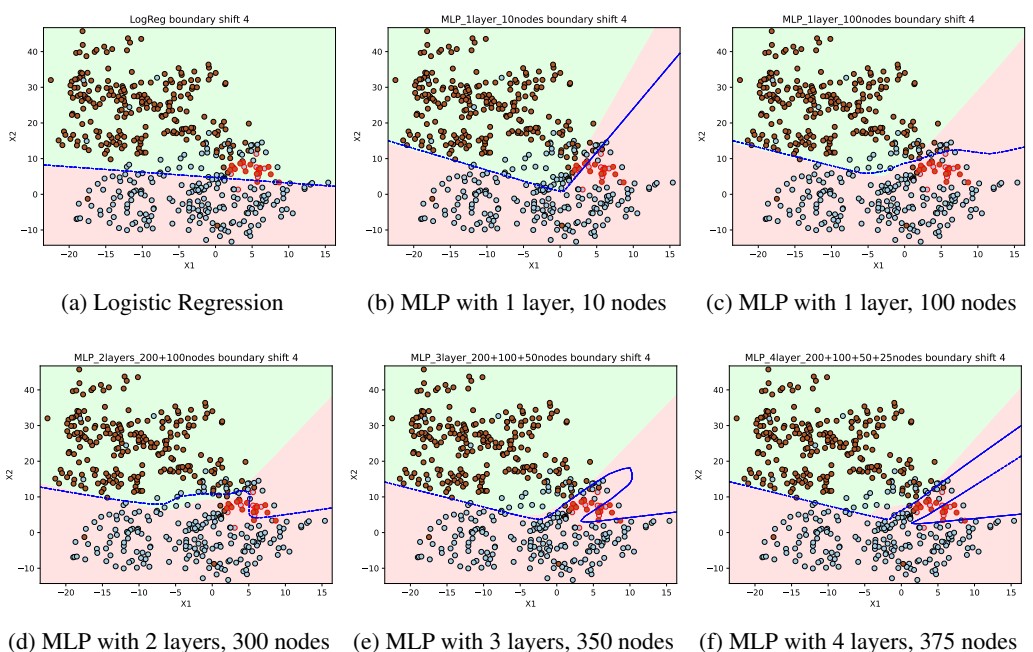

(a) Logistic Regression    (b) MLP with 1 layer, 10 nodes    (c) MLP with 1 layer, 100 nodes

(d) MLP with 2 layers, 300 nodes    (e) MLP with 3 layers, 350 nodes    (f) MLP with 4 layers, 375 nodes

Figure 6: Comparison of decision boundary shifts caused by a poisoning attack targeting subgroup 4 (red points) with $\alpha = 2.0$. The background (green-pink) represents the clean model's decision regions, while the blue line shows the boundary after the poisoning attack. The decision boundary is approximated by classifying a mesh of points across the grid. The models include Logistic Regression (a) and various Multi-Layer Perceptrons (MLP) with increasing complexity from 1 to 4 layers and varying node counts.

