# OpenReview forum: "Fragile Giants: Understanding Susceptibility of Models to Subpopulation Attacks"
_ICLR.cc/2025/Conference — Submitted to ICLR 2025_

### Official Review · Reviewer_qjmc · 2024-10-30

**Soundness:** 3
**Presentation:** 4
**Contribution:** 3
**Rating:** 8
**Confidence:** 4

**Summary:**

This manuscript inspects models' robustness against data poisoning attacks and empirically finds that models with more parameters are significantly more vulnerable to subpopulation poisoning. Fine-grained analysis suggests that attack intensity and subgroup size may also influence attack damages.

**Strengths:**

1. Clearly, a lot of effort has been put into this work. I believe the contribution is sufficient for a publication. The topic of data poisoning attacks is popular, especially in the era of generative AI.

2. The layout of the paper is clear. Most of the concepts are formally defined or introduced.

3.  The claims made are grounded by sufficient empirical evidence, as well as fine-grained analysis.

**Weaknesses:**

No efforts were made to build a defense against the attacks explored, and thus not maximizing social benevolence.

Intensity of an attack remains undefined.

**Questions:**

1. What is the intensity of a data poisoning attack? It was used in the paper but not defined/introduced.

2. It would be nice if the manuscript could talk about how the findings made in this paper can help build a defense even if the defense might not work for all subgroups.

---

> ### Author Response · Authors · 2024-11-20
>
> 1. > **Intensity of an attack remains undefined.**
>
> Thank you for pointing this out. In our paper, “intensity” refers to the poisoning ratio, denoted as $\alpha$. To avoid ambiguity, we have updated the paper to use the term “poisoning ratio” exclusively.
>
> 2. > **It would be nice if the manuscript could talk about how the findings made in this paper can help build a defense**
>
> We appreciate the suggestion and have extended the conclusion with a paragraph discussing directions towards future defenses. Our discussion highlights how our results can inform the design of defenses against poisoning attacks, emphasizing the importance of evaluating defenses across diverse subpopulations and balancing trade-offs between accuracy, fairness, privacy, and robustness.

---

> > ### Comment · Reviewer_qjmc · 2024-11-24
> > **Rebuttal acknowledged**
> >
> > i would like to thank the authors for their efforts to address my concerns. My rating stays the same (as it is acceptance).

---

### Official Review · Reviewer_yUuv · 2024-10-31

**Soundness:** 2
**Presentation:** 3
**Contribution:** 2
**Rating:** 5
**Confidence:** 2

**Summary:**

This paper primarily investigates the relationship between model complexity and vulnerability to subpopulation poisoning attacks. Through theoretical analysis and experimental research, the authors examine how machine learning models of varying complexity (such as neural networks of different sizes) respond to data poisoning attacks targeting on specific subgroups. They discover that as model complexity increases, so does the model's sensitivity to these attacks, particularly for medium-sized subgroups. The research also reveals that very small subgroups are often resistant to effective poisoning attempts.

**Strengths:**

1. Integrates theoretical framework with empirical analysis, enhancing result validity.

**Weaknesses:**

1.	While the paper identifies vulnerabilities, it does not provide possible defense strategies to mitigate the risks associated with subpopulation poisoning attacks, which could be improved.
2.	This paper touches on the impact on marginalized groups, and it could benefit from a more in-depth discussion of the ethical implications of subpopulation poisoning attacks and the responsibilities of researchers and practitioners

**Questions:**

-	Section 5.1 notes that "Small Subgroups are Difficult to Poison," which seems at odds with the conclusion suggesting defenses for subgroup vulnerabilities. The authors should clarify the relationship between subgroup size and vulnerability. The paper concludes that "Larger Subgroups Are Less Affected by Model Size" and "Model Complexity Affects Medium-Sized Subgroups Disproportionately." The authors should further analyze how these findings for medium and large subgroups differ from model performance under traditional data poisoning attacks. This would align the paper's observations with its recommendations and highlight the uniqueness of subgroup poisoning attacks.
-	I am wondering how model interpretability might influence the detection and understanding of subpopulation poisoning attacks, which could be a critical aspect of building trustworthy ML models.

**Details Of Ethics Concerns:**

This paper should discuss the ethical concerns raised from this work.

---

> ### Author Response · Authors · 2024-11-20
>
> 1. > **"Small Subgroups are Difficult to Poison," [...] seems at odds with the conclusion suggesting defenses for subgroup vulnerabilities.**
>
> Our results demonstrate that the effectiveness of subpopulation poisoning attacks varies depending on the size of the subgroup. While attacks on very small subgroups are indeed less effective—likely due to the model requiring a minimum number of points to generalize behavior across a subpopulation—attacks on larger subgroups can succeed and disproportionately impact certain subpopulations. This variation underscores the importance of developing defenses that account for subgroup-specific vulnerabilities, particularly for larger or more targeted subpopulations.
>
> 2. > **[...] how these findings for medium and large subgroups differ from model performance under traditional data poisoning attacks.**
>
> Since the poisonability of the model depends on its ability to overfit, we would expect similar model behavior for traditional poisoning attacks. However, the variance across model sizes is likely lower, because these attacks require smaller shifts in the decision boundary of the model. For example, traditional data poisoning targets a specific set of samples and typically requires significantly fewer samples to be successful because the attack only needs to adapt the decision boundary to overfit a small region. As it is generally possible even for the smaller models to perfectly memorize the full training set, we expect the difference in model size to play a smaller role. Similarly, backdoor attacks introduce behavior for samples containing an artificial trigger, making them a more distinct subpopulation and allowing model to overfit these subpopulations more easily.
>
>
> 3. > **I am wondering how model interpretability might influence the detection and understanding of subpopulation poisoning attacks.**
>
> We agree that interpretability techniques can play a crucial role in this area. By shedding light on the model's internal decision-making processes, interpretability methods can help identify irregularities or biases that suggest the presence of poisoned data affecting specific subpopulations. For example, by analyzing feature attributions or decision pathways, we might detect unexpected dependencies or anomalies when the model processes data from a targeted subgroup. Incorporating interpretability can enhance our ability to pinpoint and mitigate such attacks more effectively.

---

### Official Review · Reviewer_q8J6 · 2024-11-03

**Soundness:** 2
**Presentation:** 3
**Contribution:** 2
**Rating:** 6
**Confidence:** 3

**Summary:**

This paper explores how model complexity influences susceptibility to subpopulation poisoning attacks. The authors first prove that a learning algorithm is naturally susceptible to subpopulation poisoning attacks if it exhibits local dependence (Theorem 1). After that, the authors speculate that modern overparameterized deep learning models (e.g., MLP) also have this vulnerability since most of the existing learning algorithms have local dependency (in some settings) proved in existing works. Besides, the authors empirically verify this understanding through experiments. In particular, the authors also show that this vulnerability varies across different subgroup sizes.

**Strengths:**

1. The authors attempt to give a deeper understanding and theoretical analysis of existing attacks. It should be encouraged.
2. This is a well written paper. The definitions of symbols and the overall flow are clear.
3. The experiments are sufficient to support author’s statements to a large extent.

**Weaknesses:**

1. The scope of this paper is limited.
- In this paper, the authors focus only on the subpopulation poisoning attacks. To the best of my knowledge, this particular attack type (rather than the general data poisoning) is still not yet a widely recognized threat.
- In particular, this paper only focuses on the label flipping subpopulation poisoning attack. It further limits the generalizability of the ideas in this paper.
- The main finding (i.e., that more complex models are more vulnerable to such attacks) seems to be expected. More importantly, the authors do not provide insights on how to exploit some of the understandings found in this paper.
2. There are some potential over-claims.
- Line 19-21: To the best of my knowledge, Theorem 1 is only related to locally dependent learners instead of overparameterized models, not to mention model capacity.
- Line 41-44: missing the type of backdoor attacks [1].
- Line 130-131: please provide references or experiments to show that the previous findings are not necessarily true in subpopulation poisoning attacks.
3. Theorem 1 seems to be a straightforward extension of the one proposed in [2].
4. The authors should discuss potential applications of their findings, instead of simply highlighting the need for more attention for defenses.
5. The authors should also conduct experiments on other types of poisoning attacks, instead of just the label flipping subpopulation poisoning attack.


Minor Comments
1. There are still many typos (e.g., Line 183, Line 204).


References
1. Backdoor Learning: A Survey.
2. Subpopulation data poisoning attacks.


PS: I am not very familiar with subpopulation poisoning attacks, although I did a lot of work on data poisoning and its defenses. Please feel free to correct me if I have any misunderstanding. I am willing to increase my scores if the authors can address (parts of) my concerns.

**Questions:**

Please kindly refer to 'Weakness' for more details.

---

> ### Author Response · Authors · 2024-11-20
>
> 1. > **The scope of this paper is limited.**
>
> To clarify the rationale behind our focus, while traditional data poisoning attacks have been extensively studied, they often rely on strong assumptions—such as detailed knowledge of specific target instances or the ability to embed artificial triggers into the data. Subpopulation poisoning attacks, on the other hand, require only a general understanding of the target subpopulation, making them a more realistic and pressing threat.
> We focus on label-flipping subpopulation poisoning for two main reasons. First, it is straightforward to implement, allowing us to sidestep the unpredictability associated with more complex attack strategies while still gaining valuable insights. Second, it represents a credible threat in real-world scenarios, as it is one of the simplest methods for adversaries to execute. By focusing on this straightforward yet effective attack, we create a controlled environment to analyze how model complexity influences vulnerability.
> Our findings represent an important step toward understanding how models behave under subpopulation-targeted attacks. Specifically, we demonstrate that more complex models are more vulnerable—a phenomenon that had not been empirically validated or thoroughly analyzed before our study. We believe this provides valuable insights, paving the way for future research on protecting models against such attacks.
>
> 2. > **There are some potential over-claims.**
>
> > **Line 19-21**
>
> The use of the term "explain" in our abstract is intended to communicate that our theoretical framework provides insight into mechanisms that may contribute to the observed vulnerabilities in overparameterized models. To address this ambiguity and align with the feedback, we adapt this sentence in the revised version.
>
> > **Line 41-44**
>
> The distinction between targeted and backdoor attacks can indeed be ambiguous, as both terms have been defined differently in prior work. As outlined in the background section, we follow the definition from the original backdoor paper [A], considering backdoor attacks as orthogonal to the categorization of attacks into targeted and untargeted.
>
> > **Line 130-131**
>
> We note that this statement does not claim the findings of prior work are definitively invalid for subpopulation poisoning attacks; rather, it highlights that these findings have not yet been explicitly extended or studied in this context. To the best of our knowledge, there are no prior works that directly examine the interaction between model architecture and subpopulation poisoning attacks, as the focus of prior research has been predominantly on single-trigger backdoors or general data poisoning. The lack of references reflects the novelty of the area, not an over-claim, as the statement merely underscores the unexplored nature of subpopulation poisoning rather than making assertions about prior findings.
>
> 3. > **Theorem 1 seems to be a straightforward extension of the one proposed in [2].**
>
> Our theoretical framework builds on prior work modeling data distributions as mixtures of subpopulations [Feldman 2019, Jagielski et al. 2020]. As highlighted in Section 4.1, our theorem extends the findings of prior work by proving a broader and more generalizable result: that every subpopulation is vulnerable to poisoning, not just some subpopulations, as previously shown. This extension establishes a more comprehensive understanding of the vulnerabilities inherent in these systems, which we further analyze empirically to provide practical insights into subpopulation-targeted attacks.
>
> 4. > **The authors should discuss potential applications of their findings [...]**
>
> We agree that the paper could benefit from a discussion on potential applications. We have added a paragraph addressing this in the revised version. Our discussion highlights how our results can inform the design of defenses against poisoning attacks, emphasizing the importance of evaluating defenses across diverse subpopulations and balancing trade-offs between accuracy, fairness, privacy, and robustness.
>
> 5. > **The authors should also conduct experiments on other types of poisoning attacks**
>
> We appreciate the suggestion to explore other types of poisoning attacks. For subpopulation-specific attacks, however, the range of attack types is naturally limited by the focus on targeting identifiable subpopulations. While latent space subpopulation attacks could potentially achieve higher success rates by exploiting the model’s internal representations, they are less realistic as they rely on subgroups that do not correspond to real-world annotations and assume a stronger adversary. For more general poisoning attacks, a substantial body of prior work already exists. Our aim was to specifically focus on subpopulation poisoning to gain a deeper understanding of this realistic yet underexplored threat model.
>
> [A] BadNets: Evaluating Backdooring Attacks on Deep Neural Networks

---

> > ### Comment · Reviewer_q8J6 · 2024-11-23
> >
> > 1. Please provide some specific threat scenarios to confirm that such attacks are realistic. For me, knowing distribution of the target subpopulation is more difficult.
> >
> > 2. The main finding (i.e., that more complex models are more vulnerable to such attacks) seems to be expected. More importantly, the authors do not provide insights on how to exploit some of the understandings found in this paper.
> >
> > I am willing to increase my score if the authors can address these concerns.

---

> > > ### Author Response · Authors · 2024-11-24
> > >
> > > Thanks for following up with your questions! Here are our clarifications.
> > >
> > > 1. > **threat scenarios**
> > >
> > > Subpopulation poisoning attacks are similar to traditional data poisoning but operate under weaker assumptions about the adversary's capabilities. In standard data poisoning, an adversary usually needs detailed knowledge about specific samples or small data subsets to effectively disrupt the learning process. This limits their impact to localized effects on the targeted samples.
> > >
> > > In contrast, subpopulation poisoning exploits broader patterns within the data. The attacker only needs to identify or gather samples belonging to a specific subgroup they want to target. For example, imagine an organization collecting a massive dataset to train an image classification model, like a facial recognition system. An attacker could focus on a particular demographic subgroup—say, individuals from a minority ethnicity—and corrupt data points within that group. Instead of requiring detailed info about specific samples, the attacker uses obvious characteristics of the subgroup, such as skin color, to inject poisoned samples into the training set. Finding or creating these samples is often straightforward since it just involves recognizing or generating images that match the subgroup's features.
> > >
> > > This makes subpopulation poisoning attacks especially harmful. By leveraging demographic patterns, attackers can introduce systemic biases into the model that might go unnoticed, posing significant risks. Unlike traditional poisoning attacks that rely on specific knowledge of individual samples or victims, subpopulation poisoning uses broader population traits, allowing the attacker to affect subgroups easily. If you find it helpful, we would be happy to discuss a concrete example of such an attack in the paper.
> > >
> > > 2. > **Exploiting the understandings found in this paper**
> > >
> > > Our findings provide valuable insights into understanding model behavior through the lens of local dependence and data distribution characteristics, which can help inform the development of future defenses.
> > >
> > > One key direction we highlight in this work is the need for the development of methods and proxies to measure local dependence in large models. Such tools would enable precise assessments of a model’s susceptibility to vulnerabilities that disproportionately affect specific subpopulations within the data. Establishing robust methods for detecting and mitigating local dependence is, in our view, a crucial and underexplored area of future research that could significantly enhance model reliability.
> > >
> > > Furthermore, our results reveal a meaningful connection between subpopulation robustness and other critical properties, such as fairness and privacy. This observation raises essential questions for further exploration: Are the subpopulations at risk for robustness issues the same as those vulnerable to fairness or privacy concerns, or do these challenges affect distinct groups? Understanding these relationships is crucial for developing holistic mitigation strategies.
> > >
> > > Finally, our findings have significant implications for the interplay between subpopulation vulnerabilities and mitigation strategies. For example, can vulnerabilities related to robustness, fairness, and privacy be addressed collectively, or do inherent tensions exist between these properties? Prior work has highlighted trade-offs, particularly between privacy and fairness, which underscores the need for a nuanced approach to these challenges. Our study contributes to this discussion by emphasizing the importance of carefully balancing trade-offs between robustness, fairness, and accuracy when designing countermeasures. Addressing these trade-offs is essential for ensuring that defenses do not exacerbate other vulnerabilities, thereby achieving more meaningful improvements to model behavior.

---

> > > > ### Comment · Reviewer_q8J6 · 2024-11-24
> > > >
> > > > Thank you for your further responses. Your rebuttal has addressed most of my concerns. Although my concern regarding the scope of this paper remains to some extent, I decided to increase my score to 6 since I think we should always encourage papers to provide a deeper understanding of new things.

---

### Official Review · Reviewer_5ydg · 2024-11-04

**Soundness:** 3
**Presentation:** 3
**Contribution:** 3
**Rating:** 6
**Confidence:** 3

**Summary:**

This paper examines the vulnerability of machine learning models of various sizes to subpopulation poisoning attacks. The authors develop a theoretical framework to explain why overparameterized models are particularly susceptible to these attacks. They then conduct extensive experiments across multiple models and datasets, showing that more complex models are indeed more vulnerable to subpopulation poisoning. Additionally, the paper highlights the challenges in developing effective defenses to mitigate these specific vulnerabilities.

**Strengths:**

- The topic of understanding subpopulation attacks is interesting.
- The authors provide valuable insights, such as the finding that larger models are more susceptible to subpopulation attacks.
- Extensive experiments strengthen the credibility of the conclusions.

**Weaknesses:**

- The definition needs more illustration.
- The generalizability of the conclusions is not entirely clear.

**Questions:**

Definition 2 needs more detailed explanation. Specifically, what does f_p represent here? Also, what does S_p signify? The authors should clarify these terms when they are introduced to improve readability.

The paper mentions two types of subpopulation definitions: one based on clustering of samples’ latent space representations, and another based on predefined semantic annotations in the dataset. In their evaluation, the authors use subpopulations defined by manual annotations that provide semantic information about the samples. I would like to know if this selection might influence the conclusions—specifically, can the findings in this paper generalize to the first type of subpopulation definition? Additional insights on this point would be helpful.

Another concern is related to generalization. I noticed that the authors adopt a straightforward implementation of subpopulation attacks by flipping labels within the subpopulation. Have the authors tried other, potentially stronger, attack methods? If so, would these different attack types affect the conclusions?

---

> ### Author Response · Authors · 2024-11-20
>
> 1. > **Specifically, what does f_p represent here? Also, what does S_p signify?**
>
> f_p represents the poisoned model, i.e., the model when trained on data that includes poisoned samples, and S_p represents the indices of samples in D whose labels are flipped. Thank you for the question, we realize this can be made clearer and have adapted Definition 2 to reflect this.
>
> 2. > **can the findings in this paper generalize to the first type of subpopulation definition?**
>
> The findings in this paper can partially generalize to the clustering-based subpopulation definition, but there are important caveats. Clustering-based subpopulations often yield higher target damage because the subgroups are defined based on their separation in the model’s latent space, rather than being semantically meaningful in the real world. This results in learning behavior that is more locally dependent for those subgroups, allowing poisoning attacks to more easily influence the decision boundary for the entire subgroup, leading to higher attack success.
>
> While we would still expect similar trends—such as larger, overparameterized models being more vulnerable due to their flexibility in adapting the decision boundary—our methodology is not directly applicable. Clustering-based attacks require the adversary to generate subgroups locally in the latent space, and the effectiveness of this approach depends on whether different models adopt similar latent space representations. Misaligned representations between models could result in subgroups that are consistent for one model but not for another, introducing variability that does not arise in our manually defined, semantically meaningful subpopulations.
>
>
> 3. > **Have the authors tried other, potentially stronger, attack methods? If so, would these different attack types affect the conclusions?**
>
> We focused on label-flipping attacks because they allow us to sidestep the unpredictability associated with more complex attack strategies and because they represent a credible threat in real-world scenarios, as they are one of the simplest methods for adversaries to execute. Clustering-based subpopulation definitions could lead to stronger attacks due to their reliance on the model’s latent space, but we expect the trends to remain similar (see 2. above). Likewise, stronger poisoning attacks, such as traditional data poisoning or backdoor attacks, would likely yield comparable conclusions. However, as these attacks require significantly fewer poisoned points to achieve only minor shifts in the decision boundary, the variance in poisonability across model sizes is likely much smaller, as the flexibility of larger models becomes less critical for success when even smaller models can overfit these poisoning objectives.

---

> > ### Comment · Reviewer_5ydg · 2024-11-24
> > **Thank you for the response**
> >
> > Thank you for your response. However, I still have concerns regarding the generalizability of the findings. I agree that label-flipping is the simplest method and can produce obvious results, but it remains unclear whether the conclusion on label-flipping can be extended to more complex poisoning attacks. That said, there are some of the findings in this paper are interesting, so I maintain my borderline but positive score.

---

### Official Review · Reviewer_T5zd · 2024-11-04

**Soundness:** 2
**Presentation:** 2
**Contribution:** 2
**Rating:** 5
**Confidence:** 3

**Summary:**

The paper proposes a subpopulation poisoning attack targeting specific subgroups within data, exploiting the complexity of overparameterized machine learning models, which can inadvertently memorize and misclassify these subgroups. The paper reveals the relations of the attack success and model complexity and subgroup size.

**Strengths:**

- S1: The paper explores some key parameters of learning including model complexity and learning of similar inputs.
- S2: the paper is easy to read.

**Weaknesses:**

- W1: It is unclear how the subgroups are identified.
- W2: The significance of the work over the existing work is not clear. Also, these observations look consistent with the understanding of general supervised training.
- W3: The approach is limited to discriminative models.

**Questions:**

- Q1: Are subgroups manually identified (Line 306) or automatically clustered (Line 205)?

---

> ### Author Response · Authors · 2024-11-20
>
> 1. > **Are subgroups manually identified (Line 306) or automatically clustered (Line 205)?**
>
> Thank you for bringing this to our attention - we have updated Section 4.2 in the paper to be more explicit on this point. Subgroups in our experiments are defined based on manual annotations that reflect real-world subgroups, as this provides a realistic scenario where an adversary targets subgroups that are semantically meaningful. These annotations, also commonly used in fairness research, are readily available within the datasets we employ. For instance, in the CelebA dataset, we use attributes such as “Dark Hair,” “Young,” “Pale Skin,” “Beard,” and “Chubby” to define reasonably sized subgroups for analysis.
>
>
> 2. > **The significance of the work over the existing work is not clear [...] these observations look consistent with the understanding of general supervised training.**
>
> The consequences of learning subpopulations-specific behavior have largely been studied in the context of privacy and fairness. However, as we show in this work, it also has significant implications for robustness.
> Our work  is the first systematic exploration of the relationship between model complexity and vulnerability to subpopulation poisoning within the overparameterized regime. By evaluating attack success across models of varying sizes, we reveal that larger models, which are more prone to overfitting, are indeed more susceptible to these targeted attacks. This finding goes beyond the general understanding of supervised training by highlighting a specific vulnerability associated with model size and complexity.
> Additionally, we discuss how strategies that limit memorization—such as regularization techniques and noise addition—not only enhance privacy and fairness but also improve robustness against poisoning. However, implementing these countermeasures introduces trade-offs with accuracy and other performance aspects. Our work emphasizes the need to carefully balance these factors when designing defenses.

---

> > ### Comment · Reviewer_T5zd · 2024-11-21
> >
> > Thank you for the response. In a sense, poisoning is regular learning of the features, and also targeted attacks have been studied in previous works, which is very similar from the model perspective, such as "Detecting Backdoor Attacks on Deep Neural Networks by Activation Clustering" or "Backdoor learning curves: explaining backdoor poisoning beyond influence functions". Since it is a regular learning from the features present in the data, this is an expected result. Thus, the main message based on the response doesn't seem to make a significant contribution, and I will keep my recommendation.

---

### Official Review · Reviewer_aRUC · 2024-11-04

**Soundness:** 4
**Presentation:** 3
**Contribution:** 4
**Rating:** 8
**Confidence:** 4

**Summary:**

In this paper, the authors explore the relationship between subpopulation attacks and the complexity/overparameterization of machine learning models. Subpopulation attacks are a form of model poisoning attack in which an adversary targets a specific distribution instead of isolated samples and aims to degrade model performance on that specific distribution without significantly impacting the overall performance of the model. The authors theoretically and experimentally proved that ML models that exhibit local dependence (including larger and overparameterized models) are more susceptible to subpopulation attacks. Although the paper focuses on subpopulation attacks, I believe that this work could be helpful in improving fairness-aware ML.

**Strengths:**

1. Theoretical explanations for local dependence vs. susceptibility to subpopulation attacks are supported by experimental results.
2. The experimental setup contains different types of dataset (tabular, image, and text), which strengthens the authors' claim.

**Weaknesses:**

1. The authors consider only the binary classification case, which limits the full exploration.

**Questions:**

1. Do you think subpopulation attacks could be used to measure fairness in ML? Or are ML models that incorporate bias removal less susceptible to subpopulation attacks?
2. Is it possible to detect and/or mitigate local dependence?
3. Could one subgroup be affected by the subpopulation attack target another subgroup due to their "closeness"?
4. Page 1, line 053: repetition of "more". Page 10, line 535: missing full stop.

---

> ### Author Response · Authors · 2024-11-20
>
> 1. > **Do you think subpopulation attacks could be used to measure fairness in ML? Or are ML models that incorporate bias removal less susceptible to subpopulation attacks?**
>
> Thank you for this question. The idea of per-subgroup robustness is indeed closely related to fairness, as different models may treat subgroups disparately, leading to vulnerabilities. Prior work has highlighted the connection between per-subgroup accuracy and fairness, as well as privacy [A], suggesting that these dimensions are intertwined. Mechanisms designed to reduce bias, such as regularization and noise addition, are likely to enhance robustness to poisoning attacks. However, these countermeasures also come with tradeoffs among accuracy, privacy, robustness, and fairness. Hence, the design of mitigations requires careful balancing of all these objectives to achieve the desired outcomes.
>
> 2. > **Is it possible to detect and/or mitigate local dependence?**
>
> Detecting and mitigating local dependence is a challenging yet important problem that warrants further investigation. At present, there are no established methods to empirically measure the degree of local dependence ($\delta$) for a given neural network architecture and specific subpopulations. Developing such methods would be highly valuable, as they could provide precise assessments of how susceptible a model is to vulnerabilities affecting certain groups within the data.
>
> To detect local dependence, one could explore analyzing the model's performance across different subpopulations to identify inconsistent behaviors or biases. Techniques from interpretable machine learning, such as examining feature attributions or conducting sensitivity analyses, might offer insights into how strongly the model relies on local patterns within specific subgroups.
>
> Mitigation strategies could involve designing regularization techniques that explicitly penalize excessive reliance on local features. For instance, incorporating penalties that encourage the model to consider broader contexts or integrating data augmentation methods to reduce overfitting to local patterns might help. However, it's important to recognize that some level of local dependence is often necessary for capturing essential patterns in the data, and overly aggressive mitigation could harm the model's overall performance.
>
> We recognize the significance of this issue and believe that developing robust methods to detect and mitigate local dependence is a crucial direction for future research. Advancements in this area would enhance our ability to build models that are not only accurate but also fair and reliable across different subpopulations.
>
>
> 3. > **Could one subgroup be affected by the subpopulation attack target another subgroup due to their "closeness"?**
>
> Yes, it is entirely possible for one subgroup targeted by a subpopulation attack to inadvertently affect another subgroup due to their proximity. Since our subgroup definitions do not rely on any explicit closeness metric, samples from different subgroups can be arbitrarily close in the feature space. As a result, the shift in the decision boundary caused by the attack may spill over and impact neighboring subgroups. This effect is also illustrated in some of the visualizations in Figure 6, where the influence of the attack extends beyond the targeted subgroup.
>
> [A]  Bagdasaryan et al., “Differential Privacy Has Disparate Impact on Model Accuracy”

---

> > ### Comment · Reviewer_aRUC · 2024-11-24
> > **Official comment by Reviewer aRUC**
> >
> > Thank you for your responses to me and all other reviewers. My rating stays the same since it is acceptance.

---

### Official Review · Reviewer_XPkD · 2024-11-04

**Soundness:** 2
**Presentation:** 3
**Contribution:** 2
**Rating:** 5
**Confidence:** 3

**Summary:**

This study highlights that overparameterized models are particularly vulnerable, often failing to detect issues in smaller, interpretable subpopulations. The analysis reveals a strong relationship between model complexity and susceptibility to such attacks, exacerbated by the long-tailed nature of modern datasets. These findings stress the need for subpopulation-specific defenses, as traditional approaches may be insufficient for increasingly complex systems.

**Strengths:**

1.	The study provides a robust theoretical framework that highlights the vulnerability of locally-dependent mixture learners to subpopulation poisoning attacks. This builds on existing knowledge of how long-tailed data distributions are memorized, offering a deeper understanding of the challenges in defending against these attacks.
2.	The research empirically demonstrates that complex models exhibit significant shifts in their decision boundaries when exposed to subpopulation poisoning. This finding highlights the vulnerability associated with increased model complexity.
3.	The study conducts an extensive empirical analysis of realistic, overparameterized models across diverse real-world image and text datasets. By executing 1,626 individual poisoning attacks on various combinations of dataset, subpopulation, model, and parameters, it robustly establishes that larger models are more susceptible to subpopulation poisoning attacks.

**Weaknesses:**

1. The paper's novelty is unclear, as prior research, such as that by Jagielski et al., has proposed two methods for defining subpopulations: one based on data annotations and the other using clustering techniques. The authors then use this foundation to establish Theorem 1, which seems unchallenging.

2. The authors assert that "In this model, subpopulations have distinct supports, meaning that each data point is associated with only one subpopulation." However, in real-world datasets, some data points may belong to multiple subpopulations.

**Questions:**

1. What is the attack scenario that the adversary knows the specific subpopulation within the mixture distribution to which the validation samples belong?

2. The authors focus on binary classification. How would this approach apply to a more complex task? Would it still be effective?

3. In Figure 5b, ResNet 50 appears to be more vulnerable to subpopulation attacks than ResNet 101. Could you elaborate on the reasons for this difference?

4. The authors conduct 1,626 individual poisoning attacks across various combinations of dataset, subpopulation, model, and alpha. I’m curious if different poisoning attack methods yield varying effects on the models. Could you elaborate on that?

---

> ### Author Response · Authors · 2024-11-20
>
> 1. > **The paper's novelty is unclear, as prior research [...] has proposed two methods for defining subpopulations**
>
> Our goal is not to propose a new type attack, similar to Jagielski et al. Instead, we build upon their work to investigate the relationship between model complexity and subpopulation poisoning attacks, focusing on how attack strength varies across subpopulations and model sizes. This perspective provides new insights into the dynamics of such attacks rather than introducing new attacks.
>
>
> 2. > **The authors [...] establish Theorem 1, which seems unchallenging**
>
> Our theoretical framework builds on prior work that models data distributions as mixtures of subpopulations [Feldman 2019, Jagielski et al. 2020]. While previous research demonstrated that an attack exists for _some_ subpopulation in the case of locally dependent learners, Theorem 1 extends this by showing that _every_ subpopulation is vulnerable to poisoning, highlighting a broader and more generalizable vulnerability that we then analyze empirically.
>
>
> 2. > **in real-world datasets, some data points may belong to multiple subpopulations.**
>
> The theoretical model of mixtures of subpopulations can indeed be restrictive by assuming that each data point belongs to only a single subpopulation, i.e., that the subpopulations in the data distribution have disjoint supports. Considering the general case that allows overlap between subpopulations is significantly more complicated, but the same insight still applies: If the learner observes enough poisoned points sampled from the same subpopulation, its behavior will move towards the poisoned label for that subpopulation.
>
> 3. > Q1 **What is the attack scenario that the adversary knows the specific subpopulation [...] to which the validation samples belong?**
>
> We assume the adversary has knowledge of the target subpopulation but not the specific target samples within the validation set. This scenario appears, for example, in cases where a service provider aggregates data from multiple sources for broad deployment. For instance, in an image classification task (e.g., CelebA), an adversary might aim to poison behavior for a demographic defined by attributes like hair color, ethnicity, or gender, without knowing the exact images involved. Such an attack requires generalization across the subpopulation, making it more challenging and aligning with practical adversarial capabilities, as subpopulation-level knowledge is a weaker assumption than point-level knowledge.
>
> 4. > Q2 **The authors focus on binary classification. How would this approach apply to a more complex task?**
>
> Our approach can be extended to multiclass classification by flipping the label of poisoned points from their original class $c$ to a target label $t$, where $t \neq c$. While the methodology remains largely unchanged, this extension requires thoughtful consideration of the target label. Specifically, $t$ should either be chosen in a meaningful way or determined by iterating over all possible $t \neq c$ to understand its impact on the poisonability of the subpopulation.
>
> 5. > Q3 **ResNet 50 appears to be more vulnerable to subpopulation attacks than ResNet 101**
>
> Indeed, Figure 5b shows that ResNet50 exhibits comparable and occasionally slightly higher target damage than ResNet101. This likely reflects the convergence of poisonability for larger models, as highlighted by Figures 2b and 2c, where target damage stabilizes across the largest models in both datasets. Notably, ResNet18 remains significantly less poisonable than both ResNet50 and ResNet101, underscoring the role of model size and complexity in determining vulnerability to subpopulation attacks.
>
>
> 6. > Q4 **I’m curious if different poisoning attack methods yield varying effects on the models**
>
> Indeed, different poisoning attack methods are likely to yield varying effects on models due to differences in their objectives and mechanisms. For instance, traditional data poisoning attacks typically target a specific set of samples and require relatively few poisoned points (i.e., less than 50) to succeed, as they aim to shift the decision boundary to overfit a small region. As even smaller models can perfectly memorize the full training set, the variance in poisonability across model sizes is expected to be lower for these attacks compared to subpopulation poisoning. Similarly, backdoor attacks rely on modified samples with artificial triggers, which create a distinct subpopulation. This distinctiveness reduces their dependence on model size, as even less complex models can effectively overfit on the backdoor samples.

---

> > ### Comment · Reviewer_XPkD · 2024-11-25
> >
> > "Different poisoning attack methods are likely to yield varying effects on models due to differences in their objectives and mechanisms.  This distinctiveness reduces their dependence on model size, as even less complex models can effectively overfit on the backdoor samples." As a result, it becomes challenging to understand how the strength of an attack varies across different subpopulations and model sizes.

---

### Author Response · Authors · 2024-11-20

We thank the reviewers for their constructive feedback, which has greatly enhanced the quality of our paper. Below, we provide detailed, point-by-point responses to each comment, accompanied by the revised manuscript. The updated version includes an expanded discussion on how our results contribute to and inform the development of defenses, along with writing improvements for better presentation. All revisions discussed in the responses are clearly highlighted in blue for your convenience.

We also provide an anonymized link to a `latexdiff` PDF highlighting all changes of our revision: https://anonymous.4open.science/r/iclr-robustness-2025-DAF7/revision_1_diff.pdf

---

### Meta-Review · Area_Chair_uvWN · 2024-12-11

**Metareview:**

This paper studies the vulnerabilities of models with different complexity to subpopulation poisoning attacks. With the theoretical frameworks they developed and experiments on MLP, BERT, and ResNet, the authors claim that increasing complexity makes models more sensitive to subpopulation attacks. Although some reviewers appreciate this work's theoretical results and potential impacts on fairness, most reviewers are still concerned about the technical novelty of this paper's theoretical novelty, scope, and generalization of the paper's claims. Therefore, I think this paper still needs modifications before its acceptance.

Strengths:

1. The theory and experiment in this paper can corroborate each other.

2. Experiments on different modalities are included to support the paper's claim

Weaknesses:

1. The scope of this paper is somehow limited. This paper only conducts experiments on limited poison attack methods, not to mention that subpopulation is only a small part of data poisoning attacks. During the rebuttal, the authors also agreed that "Different poisoning attack methods are likely to yield varying effects on models due to differences in their objectives and mechanisms." Therefore, I wonder whether the paper's analysis is limited to some subpopulation attacks. Therefore, I do not recommend it to be accepted.

2. The empirical experiments are limited to support their results on model complexity. As model complexity is not only the model size and depth, model structures, activation method, and width should also be included. Although this paper is already BERT, MLP, and ResNet, these are studied separately on different datasets. Therefore, current conclusions on model complexity are somehow over-claimed. I recommend the authors add more model types, widths, and activations for different datasets, for example, ViT, ResNeXT, and WideResNet can be included in CelebA's experiments.

**Additional Comments On Reviewer Discussion:**

The reviewers and authors have an active discussion in the rebuttal period. After the rebuttal, most reviewers still have concerns about the paper's novelty and scope.

---

### Decision · Program_Chairs · 2025-01-22

Reject